# Application of Granular and Non-Granular Organic Fertilizers in Terms of Energy, Environmental and Economic Efficiency

**Egidijus Šarauskis** [1][ID]**, Vilma Naujokienė** [1][ID]**, Kristina Lekavičienė** [1,*][ID]**, Zita Kriaučiūnienė** [2]**, Eglė Jotautienė** [1]**, Algirdas Jasinskas** [1][ID] **and Raimonda Zinkevičienė** [1]

1   Institute of Agricultural Engineering and Safety, Agriculture Academy, Vytautas Magnus University, Studentu Str. 15A, LT-53362 Akademija, Lithuania; egidijus.sarauskis@vdu.lt (E.Š.); vilma.naujokiene@vdu.lt (V.N.); egle.jotautiene@vdu.lt (E.J.); algirdas.jasinskas@vdu.lt (A.J.); raimonda.zinkeviciene@vdu.lt (R.Z.)

2   Institute of Agroecosystems and Soil Sciences, Agriculture Academy, Vytautas Magnus University, Studentu Str. 11, LT-53361 Akademija, Lithuania; zita.kriauciuniene@vdu.lt

\*   Correspondence: kristina.lekaviciene@vdu.lt

**Abstract:** Granular organic fertilizers have been increasingly used in agriculture due to the longer delivery of nutrients to plants and the milder impact on the environment. The aim of this study was to determine the energy, environmental and economic efficiency of granular and non-granular organic fertilizers. Four technological scenarios of organic fertilizer use were used for comparative assessment: (1) manure fertilization (16.0 t ha$^{-1}$), (2) manure fertilization (30.0 t ha$^{-1}$), (3) manure pellet fertilization (2.0 t ha$^{-1}$), and (4) fertilization with meat and bone meal pellets (0.7 t ha$^{-1}$). Experimental studies using the mass flow method of laser spectroscopy were performed to evaluate the comparative environmental impact of granular and non-granular organic fertilizers. Economic assessment was performed for mechanized technological operations of loading, transportation and distribution of organic fertilizers, estimating the price of aggregates used and fuel consumed, the costs of individual technological operations and other indirect costs. The results showed that for mechanized technological operations, when fertilizing with granular organic manure and meat and bone meal fertilizer, energy consumption is 3.2 to 4.0 times lower compared to fertilization with manure. The average ammonia (NH$_3$) emissions from granular organic fertilizers were found to be six times lower than from non-granular organic fertilizers. The lowest costs for mechanized works were incurred when using meat and bone meal pellets, the highest economic benefits of organic fertilizers by elements was when using manure 30 t ha$^{-1}$, and the highest costs for organic fertilizers were incurred when using manure pellets.

**Keywords:** pellets; manure; emissions

## 1. Introduction

In recent decades, enhancing agricultural productivity without harming the environment has become an increasingly important issue as the population grows [1]. In order to obtain an optimal yield, it is crucial to ensure soil health and to provide crops with the necessary nutrients. By maintaining a nutrient balance, nutrients that have been consumed by pre-crop plants can be returned to the soil with the help of fertilizers [2–4]. Reference [5] states that proper crop fertilization can increase crop production by as much as 30–50%. Crop cultivation technologies differ, as do nutrient needs. These needs are met by fertilizers with known levels of nutrients that can accelerate plant growth and assure higher yields [6]. In the case of organic fertilizers, up to 35 tonnes of manure per hectare can be used to meet the rates of nitrogen (N) and phosphorus (P) fertilizers, assuming a content of about 4.8 kg N t$^{-1}$ and 1.5 kg P t$^{-1}$ [7].

Intensive agricultural production can have a significant negative impact on the environment [8]. Extensive fertilization with both mineral and organic fertilizers and the application of conventional fertilization methods at a uniform rate have negative effects

on the soil, environment and human health [9,10]. Reducing the overuse of chemical fertilizers in crop production is therefore a key factor in assuring healthier soils, healthier food, and more economical, efficient and cleaner agricultural production [11]. Unbalanced quantities of nutrients in organic and mineral fertilizers during and immediately after the fertilization process are dispersed in the environment and have a negative impact on climate change, and when released into groundwater and surface water, intensify eutrophication and acidify environmental elements [12]. Annual total agricultural emissions are around 4.6 Gt $CO_{2eq}$ [13]. As the quantity of mineral fertilizers applied increases, soil quality deteriorates, emissions of harmful gases into the air increase, as does water pollution [14]. Meanwhile, organic fertilizers supplement the soil with organic matter and essential substances for plants, such as nitrogen, phosphorus, potassium, calcium, magnesium and sulphur. These fertilizers are also a source of trace elements [15]. On the other hand, livestock manure applied to the soil surface or incorporated into the soil, as well as manure management and mineral N fertilizers, are the main sources of $NH_3$ emissions, accounting for about 52% of the total $NH_3$ emissions. The agricultural sector accounts for the largest share (92%) of $NH_3$ emissions in the European Union [16,17]. One of the most important aspects of agricultural optimization is to reduce its environmental impact, especially in terms of emissions [18]. It is very important not to spread too much fertilizer, as improper use of fertilizers reduces the efficiency of plant nutrient use and increases the overall production costs. This concept is based on the theory of precision farming. Reference [19] points out that that precision farming is associated with economically accurate farming, which requires prior knowledge of soil areas, their heterogeneity, type and history. Based on this knowledge, precision farming allows to achieve results when calculating the variable amount of fertilizer, ensures the reduction of crop growth differences in the field and the negative impact on the environment caused by leaching of too much fertilizer used for fertilization [20]. Plant nutrients that are not absorbed during vegetation significantly increase harmful gas emissions from agriculture. Excessive nitrate levels in soil due to leaching can contribute to groundwater and atmospheric pollution [21]. One way to reduce harmful gas emissions, energy consumption and expenses is to optimize the nitrogen added with fertilizers [22,23]. In addition, soil organic carbon sequestration has a great potential to reduce the negative environmental and greenhouse effects of agriculture [18].

The conversion of mineral fertilizers to organic ones is based on soil improvement and the reduction of negative environmental impacts. Rational use of fertilizers can reduce unwanted climate changes [22,24]. Reference [25] states that due to slow mineralization, organic fertilizers are leached into water more slowly than mineral fertilizers, and thus water pollution with nitrogen compounds is lower.

Livestock manure is one of the most valuable organic fertilizers, but the technological process of spreading has low productivity. In addition, manure fertilization causes significant nutrient losses. As a result, the production of granular organic fertilizers has recently become more popular, with the main aim of converting high-moisture organic matter, such as manure, manure mixtures, meat and bone waste or other organic matter, into pellets that are convenient to spread in the field [26]. The pellets are usually produced with the diameter of 4 or 6 mm so that they can be easily spread with mineral or organic fertilizer spreaders. Once in the soil, organic fertilizer pellets become wet, decompose and release nutrients [27]. Analysis of the simulation results of granular organic fertilizer spreading showed that at a feed flow of granular organic fertilizer of 200 g s$^{-1}$, the distribution of pellets with the diameter of 4 mm was more even. At a feed flow of 400 g s$^{-1}$, organic fertilizer pellets with the diameter of 6 mm were more evenly distributed [28].

Meat and bone meal is a by-product of the meat processing industry and is increasingly used in Europe as an organic fertilizer. These fertilizers contain important nitrogen, phosphorus, potassium and calcium substances [26,29,30] that are valuable for agricultural crops. Organic fertilizers from meat and bone meal have an indirect positive effect on the environment by limiting the demand for mineral fertilizers and providing a way to dispose of large amounts of waste from the meat processing industry [31]. Depending on the

type of the plant, the rate of granular organic fertilizer per hectare is from 1 to 2 tons [27]. Other authors who studied ungranulated meat and bone meal fertilizer used from 630 to 2530 kg ha$^{-1}$ as a nitrogen fertilizer for cereals [31].

There is quite a lot of research and scientific work on the effects of organic fertilizers on soil and plants, but there are almost no studies analyzing the differences in energy and economic indicators of technological operations of the use of granular and non-granular organic fertilizers. It is also not clear how the environmental impact of granular and non-granular organic fertilizers changes. Therefore, the aim of this study was to evaluate the energy and economic efficiency of technological operations of manure pellets, meat and bone meal pellets and different rates of animal manure, as well as to investigate the impact of manure and meat and bone meal pellets on the environment. This research highlights significant differences between granular and non-granular organic fertilizer, confirming the superiority of granular organic fertilization related to the environmental, energy and economic aspects. The study was chosen to evaluate possibilities by reducing environmental pollution, energy consumption and costs to obtain similar benefits from fertilizers by nutritious elements.

## 2. Materials and Methods

### 2.1. Energy Assessment of Organic Fertilizer Use

A comparative energy assessment describing the energy efficiency of granular and non-granular fertilizers was performed. The energy consumption for machinery, human labor and diesel fuel has been calculated for those technological operations that apply to organic fertilization. The main indicators of mechanized agricultural operations were selected and calculated in accordance with the recommendations prepared by the Lithuanian Institute of Agrarian Economics [32]. A self-propelled loader (Weidemann 4080T, Weidemann GmbH, Diemelsee-Flechtdorf, Germany) with a power of 65 kW was selected for organic fertilizer loading. For transportation and spreading of fertilizers was selected tractor with a power of 138 kW (John Deere 6175M, Deere & Company, Moline, IL, USA), capable of working with 16–18 tonnes organic fertilizer spreader (Rollforce 5517, Rolland, Brest, France). These machines were matched to each other according to their power and operating parameters, according to the manufacturer's recommendations and reference [32]. Useful energy consumption for different organic fertilizers was calculated assuming that the energy equivalent of human working time is 1.96 MJ h$^{-1}$, diesel fuel is 39.6 MJ L$^{-1}$ and machine use is 357.2 MJ h$^{-1}$ [33,34]. Energy equivalents of different organic fertilizers were selected according to the scientific literature [33,35,36].

In order to estimate and to compare the energy consumption of granular and non-granular organic fertilizer application technological operations, 4 technological scenarios were developed: (1) manure fertilization (16.0 t ha$^{-1}$), (2) manure fertilization (30.0 t ha$^{-1}$), (3) manure pellet fertilization (2.0 t ha$^{-1}$), (4) fertilization with meat and bone meal pellets (0.7 t ha$^{-1}$) (JSC Cignera, Kaunas, Lithuania).

### 2.2. Assessment of Environmental Impact of Organic Fertilizers

Experimental research was conducted in 2020–2021 at Vytautas Magnus University Agriculture Academy. Experimental research was performed using the mass flow method of laser spectroscopy to evaluate two scenarios for the use of organic fertilizers: first, the environmental impact of manure pellets made from cattle manure compost, and second, the environmental impact of ungranulated manure. Cattle manure, which emits large amounts of $NH_3$ into the environment, was used in the research. $NH_3$ emissions from manure studies were performed on an experimental study stand (Figure 1). Samples of ungranulated manure or otherwise manure pellets were discharged into a 25-L chamber 2 (Figure 1), which was placed in a wind tunnel. The wind tunnel section for storing manure was sealed with a cover 17. The air was pumped out of the wind tunnel through a duct 4 and a directional air flow was created above the layer of organic fertilizer. Diameter of the air extraction duct was 100 mm, and the length was 1500 mm. The length of the duct

to the air sampling probe was 1000 mm or 10 times its diameter. This length of the duct ensured the conditions for the establishment of a laminar air flow in it. The ventilation intensity of the wind tunnel was changed by a damper 8 installed in the air extraction duct 4, by changing the cross-sectional area of the duct and by changing the speed of the fan 9 with a frequency converter. Air samples were taken from the duct by probes 5 and a heated hose 13 and supplied to the gas analyzer 14. Air was supplied to the analyzer continuously by a pump 16 with a capacity of 6.0 L min$^{-1}$. To prevent air condensation, it was warmed in a hose 13 and heated to 150 °C in electrically heated valves 15. At the start of the study, the gas analyzer 14 was programmed to record the average $NH_3$ gas concentration values every minute. The emission intensity was calculated according to the mass flow method. The mass flow method is used to determine the total emission intensity in the tank. In this case, the ventilation intensity of the tank and the difference of the gas concentrations entering and leaving the tank are measured. The emission intensity is then calculated according to the formula:

$$E = \Delta C \cdot G \tag{1}$$

where E is the emission intensity, mg m$^{-2}$ h$^{-1}$; $\Delta C$ is the difference between the concentrations of the gases entering and leaving the room, mg m$^{-3}$; G is room ventilation intensity, m h$^{-1}$.

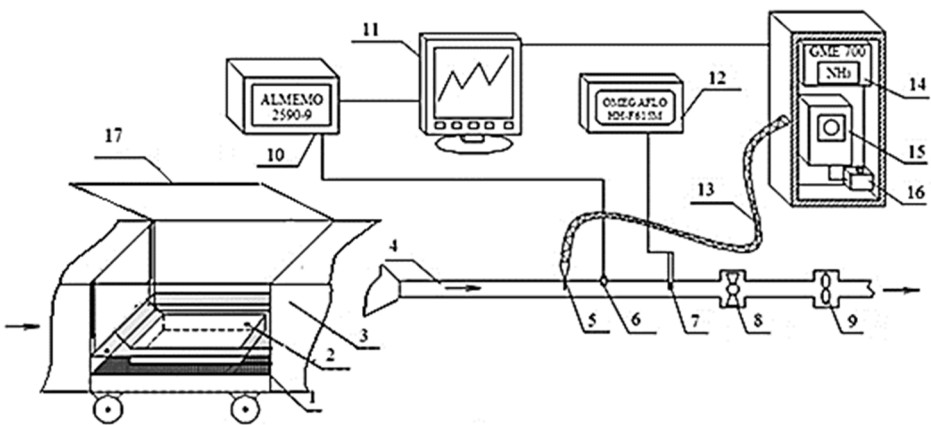

**Figure 1.** Scheme of test stand of NH3 emissions from manure: 1—pallet with wheels, on which the manure chamber is placed; 2—manure storage chamber; 3—airtight wind tunnel chamber made of transparent material; 4—air extraction duct; 5—air sampling probe; 6—thermal anemometer sensor; 7—temperature and humidity sensors; 8—valve; 9—fan with a frequency converter; 10—meter-accumulator "Almemo 2590-9" (Rottenburg, Germany); 11—computer (AMR program); 12—anemometer "OMEGAFLO HH-F615M" (Tokyo, Japan); 13—heated air supply hose; 14—laser gas analyzer "GME700" (Minnesota, United States); 15—electrically heated three-channel valves; 16—diaphragm air pump; 17—cover.

Experimental tests are performed on a laboratory stand until a crust forms on the surface of the granulated and non-granulated manure. Manure is then poured into plastic chambers. Manure is stored in both chambers under the same aerobic conditions at a constant temperature of 18.3 ± 0.2 °C. Periodically, each manure chamber is placed in a wind tunnel and $NH_3$ emissions from it are determined. The test is performed at a constant air flow above the manure and at a constant ambient temperature. The air velocity in the 100 mm diameter duct before each test is determined to be 0.1 ± 0.02 m s$^{-1}$.

A GME700 analyzer with $NH_3$ measurement ranges from 0 to 2000 ppm was used to measure $NH_3$ gas concentration. The measuring principle of this device is based on laser spectroscopy. The analyzer is equipped with heating of the gas pumped, so gas conditioning means ensure that the cell is not contaminated or condensed. Electrically heated three-channel valves are used to ensure that the operating temperature is higher

than the dew point of the sample. Operating mode is automatic—continuous or cyclic measurement with data acquisition.

Granular and non-granular manure organic fertilizers, from which the intensity of $NH_3$ emissions is determined, were poured into the chamber and spread in a uniform layer. The fan is switched on and the damper or frequency converter sets the air velocity in the duct, which is kept constant during the tests. The gas analyzer pumps are switched on as well. The gas analyzer must be switched on 1 h before starting the tests. Temperature and gas concentration measurements are then performed. The manure temperature and the ventilation intensity of the chamber are maintained constant during the tests. The tests are performed until the gas concentration values stabilize and change only slightly, the results are recorded and stored in the data storage devices.

*2.3. Economic Assessment of Organic Fertilizer Use*

Economic assessment was performed for mechanized technological operations of loading, transportation and distribution of organic fertilizers, estimating the price of aggregates used and fuel consumed, the costs of individual technological operations and other indirect costs. Expenses influencing the change of economic costs in technological operations were estimated according to the prices of mechanized agricultural services prepared by the Lithuanian Institute of Agrarian Economics [32]. According to these rates, the normative price of manure ranges from 7 to 12 EUR $t^{-1}$ (adopted 8 EUR $t^{-1}$), granular organic fertilizers (manure) from 190 to 220 EUR $t^{-1}$ (adopted 200 EUR $t^{-1}$) and granular meat and bone meal fertilizers from 350 to 430 EUR $t^{-1}$ (adopted 390 EUR $t^{-1}$). Two different manure fertilization rates were chosen for the calculations, i.e., 16 and 30 t $ha^{-1}$. Meanwhile, in accordance with the manufacturers' recommendations, the rates of pellet fertilization were chosen as follows: manure pellets—2.0 t $ha^{-1}$, and meat and bone meal pellets—0.7 t $ha^{-1}$. The prices of fertilizers nitrogen N, phosphorus $P_2O_5$ and potassium $K_2O$ were selected according to the literature [11]: for N—0.9 EUR $kg^{-1}$, for $P_2O_5$—0.8 EUR $kg^{-1}$ and for $K_2O$—0.5 EUR $kg^{-1}$, respectively.

*2.4. Statistical Data Analysis*

Experimental studies on environmental impact of non-granulated manure and of manure pellets were performed in 6 replicates separately with each type of organic fertilizer. Arithmetic averages, standard deviation and their intervals of confidence were determined with probability level ($p < 0.05$), respectively.

## 3. Results

### *3.1. Energy Assessment of Technological Operations of Organic Fertilizer Use*

A comparative analysis of the main parameters of granular and non-granular organic fertilizer loading, transportation and spreading technological operations (Table 1) showed that the working time and the fuel consumption required for manure application per hectare is about three times higher compared to spreading the pellets of manure or meat and bone meal. Evaluating the results of the analysis, it should be noted that all technological operations were assessed as a general complex because the capacity of each operation separately could be significantly higher. For example, loading capacity is about 55 t $h^{-1}$, while manure spreading field capacity is about 8.6 ha $h^{-1}$. However, when working with only one organic fertilizer spreader, part time the machine is spreading the fertilizer and part time it is transporting the fertilizer to the field. The transportation distance from the farm to the field and back was 15 km. Because of this reason, the comparative analysis per hectare is clearer when the two technological operations are combined together.

The energy consumption of mechanized technological operations for the use of organic fertilizers was calculated according to the conversion factors given in the Materials and Methods section. Assessing the energy consumption for technological operations for each type of organic fertilizer (Table 2), it was found that the largest energy consumption was for machinery and diesel fuel. The energy consumption of manure fertilization (16 t $ha^{-1}$)

for mechanized operations was 636.4 MJ ha$^{-1}$. Increasing the fertilization rate of non-granulated manure to 30 t ha$^{-1}$ increased the energy consumption by about 17%. Energy consumption for mechanized technological operations with manure pellets or meat and bone meal pellets was 3.2 to 4.0 times lower than with non-granulated manure (16 t ha$^{-1}$). Energy consumption for spreading manure pellets was about 25% higher than for meat and bone meal pellets. As a result, the rate of fertilization of meat and bone meal pellets per hectare may be lower.

**Table 1.** The main parameters of technological operations of granular and non-granular organic fertilizers.

| Technological Operation | Tractor Power (kW) | Working Width (m) | Labor (h ha$^{-1}$) | Fuel Consumption (L ha$^{-1}$) |
|---|---|---|---|---|
| Manure loading into an organic fertilizer spreader | 65 | - | 0.67 | 0.80 |
| Manure transportation and spreading (16 t ha$^{-1}$) | 138 | 12–14 | 0.33 | 6.20 |
| Manure transportation and spreading (30 t ha$^{-1}$) | 138 | 12–14 | 0.44 | 7.90 |
| Total (16 t ha$^{-1}$): | - | - | 1.00 | 7.00 |
| Total (30 t ha$^{-1}$): | - | - | 1.11 | 8.70 |
| Manure pellet loading into an organic fertilizer spreader | 65 | - | 0.11 | 0.13 |
| Manure pellet transportation and spreading (2.0 t ha$^{-1}$) | 138 | 12–14 | 0.14 | 2.63 |
| Total: | - | - | 0.25 | 2.76 |
| Loading of meat and bone meal pellets into an organic fertilizer spreader | 65 | - | 0.04 | 0.05 |
| Meat and bone meal pellet transportation and spreading (0.7 t ha$^{-1}$) | 138 | 12–14 | 0.13 | 2.44 |
| Total: | - | - | 0.17 | 2.49 |

**Table 2.** Energy inputs of mechanized technological operations for the application of granular and non-granular organic fertilizers.

| Organic Fertilizers (Fertilization Rate) | Energy Inputs, MJ ha$^{-1}$ | | | Total |
|---|---|---|---|---|
| | Labor | Fuel | Machinery | MJ ha$^{-1}$ |
| Manure (16 t ha$^{-1}$) | 1.96 | 277.2 | 357.2 | 636.4 |
| Manure (30 t ha$^{-1}$) | 2.18 | 344.5 | 396.5 | 743.2 |
| Manure pellets (2.0 t ha$^{-1}$) | 0.49 | 109.3 | 89.3 | 199.1 |
| Meat and bone meal pellets (0.7 t ha$^{-1}$) | 0.33 | 98.6 | 60.7 | 159.6 |

In order to estimate the total energy consumption of organic fertilizers, it is also necessary to include the energy consumption of the fertilizer itself. The energy consumption of different organic fertilizers is calculated according to the equivalents published or calculated in scientific literature [33,35,36]. Table 3 shows the total energy consumption for different granular and non-granular fertilizers per hectare. The results obtained showed that the lowest energy consumption was using meat and bone meal pellets when the total energy consumption for fertilizers and mechanized technological operations was about 4136 MJ ha$^{-1}$. The highest energy consumption (14,843.2 MJ ha$^{-1}$) was for manure application with the rate of 30 t ha$^{-1}$. In this study, fertilization with manure pellets or meat and bone meal pellets consumed only 33.28 and 27.86% of energy for fertilizers and technological operations, respectively, compared to manure fertilization (30 t ha$^{-1}$).

### 3.2. Environmental Impact Assessment

Laboratory studies investigated the impact of granular and non-granular organic fertilizers on the environment by measuring the concentrations of harmful NH$_3$ gases with a specialized analyzer operating on the principle of laser spectroscopy (Figure 2). After estimating the NH$_3$ gas concentrations of granular organic fertilizers, the configuration of parts per million units of NH$_3$ concentration parameter was performed and NH$_3$ gas emissions were calculated using the mass flow method (Figure 3). After evaluating the dependence of NH$_3$ gas concentration and emissions on the storage time of pellets (minutes, hours (Figure 4), days (Figure 5)), a comparative assessment of the environmental impact of granular and non-granular organic fertilizers was performed (Figures 6 and 7).

**Table 3.** Total energy inputs of granular and non-granular organic fertilizers per hectare.

| Organic Fertilizers (Fertilization Rate) | Energy Equivalent, MJ kg$^{-1}$ | Energy Inputs for Fertilizers, MJ ha$^{-1}$ | Energy Inputs for Technological Operations, MJ ha$^{-1}$ | Total, MJ ha$^{-1}$ | Percentage, % |
|---|---|---|---|---|---|
| Manure (30 t ha$^{-1}$) | 0.47 | 14,100 | 743.2 | 14,843.2 | 100.00 |
| Manure (16 t ha$^{-1}$) | 0.47 | 7520 | 636.4 | 8156.4 | 54.95 |
| Manure pellets (2.0 t ha$^{-1}$) | 2.37 * | 4740 | 199.1 | 4939.1 | 33.28 |
| Meat and bone meal pellets (0.7 t ha$^{-1}$) | 5.68 ** | 3976 | 159.6 | 4135.6 | 27.86 |

* 2.02 MJ kg$^{-1}$ (organic fertilizers) + 0.35 MJ kg$^{-1}$ (granulation) = 2.37 MJ kg$^{-1}$; ** 5.33 MJ kg$^{-1}$ (meat and bone meal fertilizers) + 0.35 MJ kg$^{-1}$ (granulation) = 5.68 MJ kg$^{-1}$ [33,35,36].

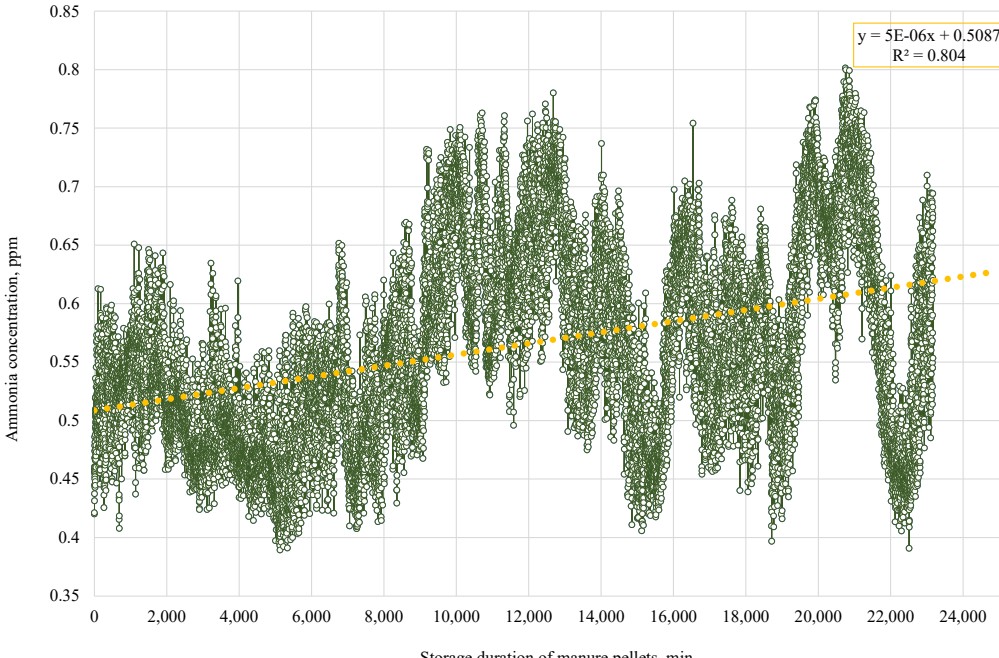

**Figure 2.** Dependence of the average NH$_3$ concentration on the storage duration of manure pellets.

It was found that after the storage of manure pellets for several weeks, the average NH$_3$ concentration changes only slightly—around 0.38–0.82 ppm. The lowest values of NH$_3$ gas concentration were determined in the average periods of 5000, $1.5 \times 10^4$, $1.85 \times 10^4$ and $2.25 \times 10^4$ min. The highest values of NH$_3$ gas concentration were determined in the average periods of $1 \times 10^4$, $1.25 \times 10^4$, $2 \times 10^4$ and $2.11 \times 10^4$ min. Despite the intensive variation, distribution and scattering of the values of NH$_3$ gas concentration in the total fixed range of values, the dependence of NH$_3$ concentration on the storage duration of the pellets (min) can be seen. A chronic gradual increase in NH$_3$ concentration from granular fertilizers according to the linear dependence on the storage time of pellets is observed.

NH$_3$ gas emissions from granular organic fertilizers were calculated using the mass flow method (Figure 3).

By recording the average NH$_3$ gas emission values from granular organic fertilizers every minute, the obvious variation of the values in the range from 0 to $2.4 \times 10^4$ min was observed. NH$_3$ gas emission values ranged from 0.09 to 0.21 mg m$^{-2}$ h$^{-1}$. The lowest NH$_3$ gas emission values were determined in the average periods of 5000, $1.5 \times 10^4$, $1.85 \times 10^4$ and $2.25 \times 10^4$ min. The highest NH$_3$ gas emission values were determined in the average periods of $1 \times 10^4$, $1.25 \times 10^4$, $2 \times 10^4$ and $2.11 \times 10^4$ min. Scatter of the points of fixed values is significant, therefore, after the correlation and regression analysis, a model for predicting the linear dependence of NH$_3$ emissions on the storage duration of pellets was derived. A monotonic trend of weak linear increase in NH$_3$ emissions was observed from the storage duration from the pellets.

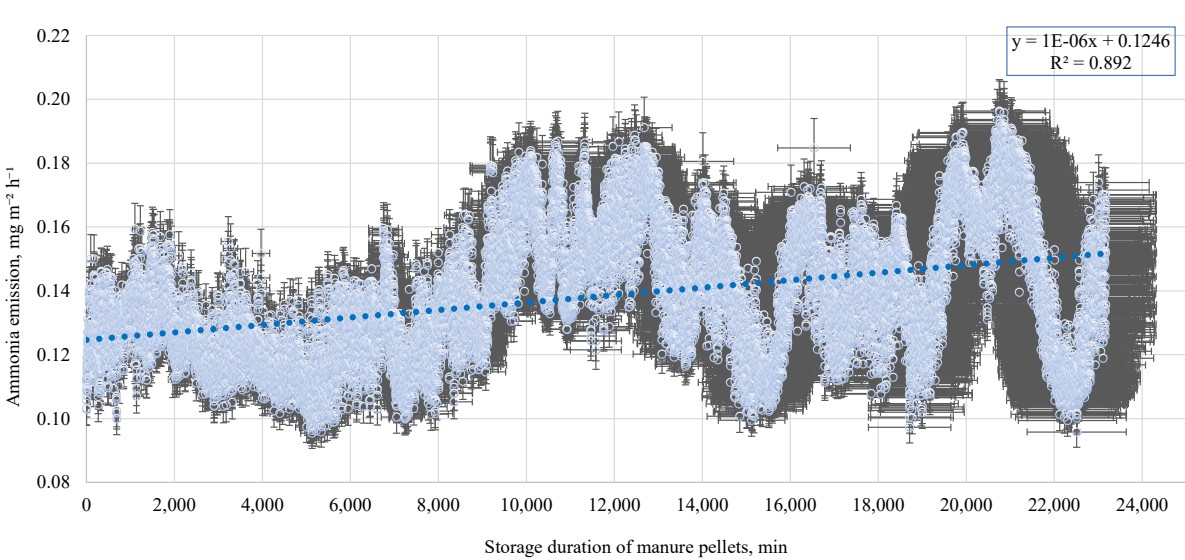

**Figure 3.** Dependence of the average NH$_3$ emissions on the storage duration of manure pellets.

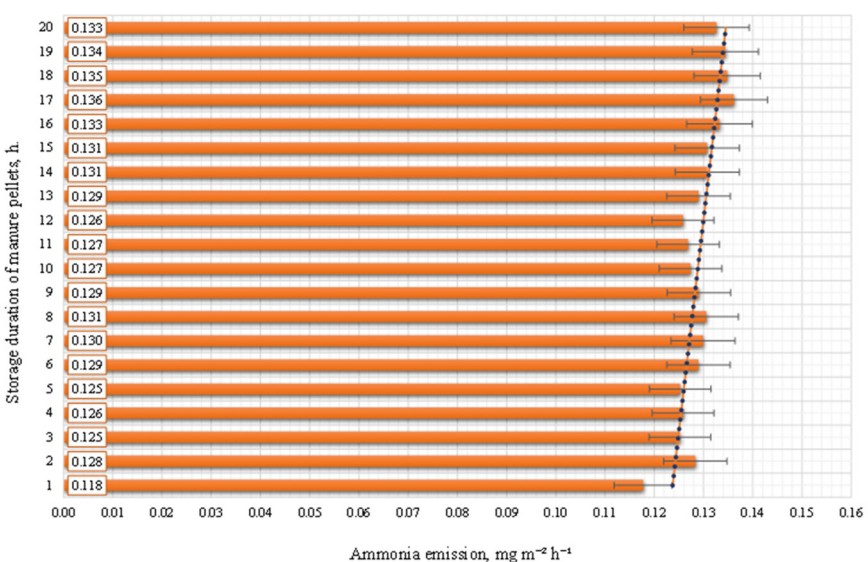

**Figure 4.** Dependence of the average NH$_3$ emissions on the storage duration of manure pellets (in hours).

After evaluating the dependence of the average NH$_3$ emissions on the storage duration of manure pellets in hours, a slight variation of the average NH$_3$ emission values was found. The lowest NH$_3$ emission values were recorded in the first hour studied, and a significant increase in NH$_3$ emissions was recorded in the second hour, which was pulsating repeated after 6, 7, 8, 9, 14, 16, 17, 18 and 19 h (Figure 4).

As found in previous studies, an upward trend in NH$_3$ emissions as a function of storage duration of granular fertilizers has been reaffirmed. It is likely that when granular fertilizer contacts the external environment and absorbs ambient moisture due to the ongoing surface reaction of other pellets with air, higher NH$_3$ gas emissions are observed if the pellets are kept in contact with the environment for a longer time period. The lowest average NH$_3$ emissions from pellets was recorded after 17, 4, 11 and 16 days. Higher intensification of NH$_3$ emissions from granular organic fertilizers was recorded after 7, 8, 9, 10, 12, 14 and 15 days of experimental studies (Figure 5).

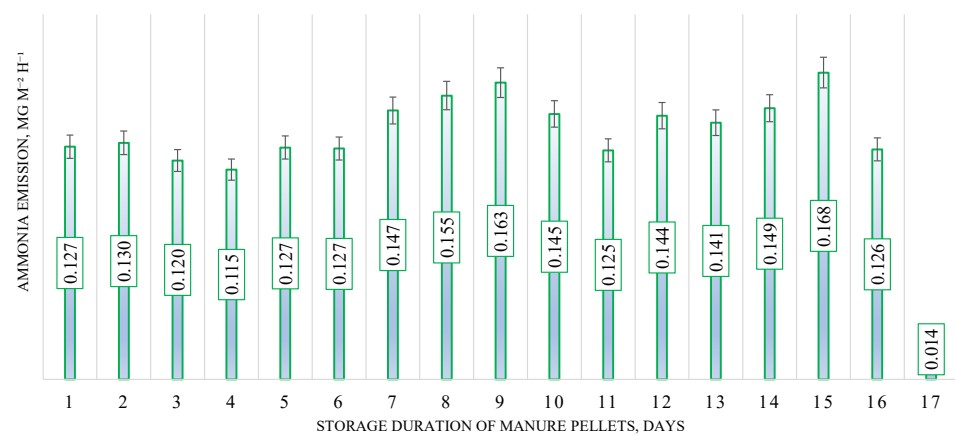

**Figure 5.** Dependence of the average NH$_3$ emissions on the storage duration of manure pellets (in days).

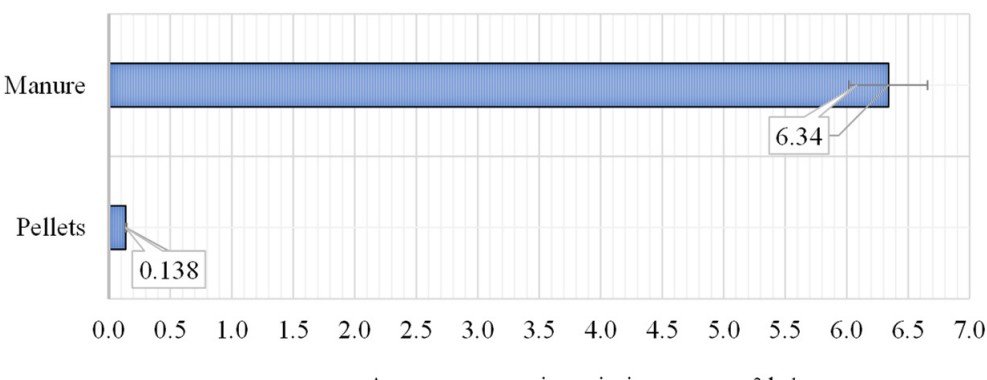

**Figure 6.** Average NH$_3$ emissions from non-granular organic fertilizers (manure) and granular organic fertilizers.

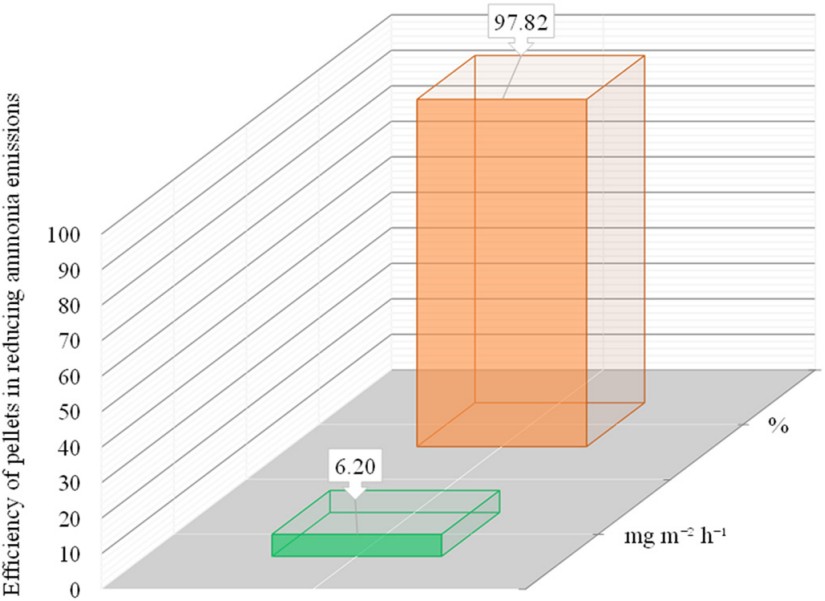

**Figure 7.** Efficiency of granular organic fertilizers (pellets) in reducing NH$_3$ emissions.

After examining the environmental impact of granular organic fertilizers in laboratory studies, a parallel study was performed by analysing the environmental impact of conventional non−granular organic fertilizers (manure) to compare and to evaluate the efficiency of the pellets (Figures 6 and 7).

The average $NH_3$ emissions from non−granular organic fertilizers (manure) and granular organic fertilizers for all study periods was summarized. The comparison showed that the average $NH_3$ emissions from granular organic fertilizers was significantly lower. The average $NH_3$ emissions from granular organic fertilizers (0.138 mg m$^{-2}$ h$^{-1}$) is more than six times lower than from non-granular organic fertilizers (manure) (6.34 mg m$^{-2}$ h$^{-1}$).

After summarizing the principle of laser spectroscopy using the mass flow method, evaluating different variants of organic fertilizers, the efficiency of granular organic fertilizers in reducing $NH_3$ emissions was determined—6.20 mg m$^{-2}$ h$^{-1}$. After summarizing the efficiency of granular organic fertilizers in reducing $NH_3$ emissions and environmental impact, it was found that granular organic fertilizers are 97.82% more effective than non-granular organic fertilizers in reducing $NH_3$ emissions. Considering the research results, it can be stated that it is important to perform fertilization with granular organic fertilizers.

Other research has shown that the use of granular biochar on manure surfaces is as effective as the use of powder to protect farmers and animals from excessive exposure to $H_2S$ and $NH_3$. After evaluating the pellets and powders by mixing pig manure for 3 h for their efficiency in reducing gas, the pellets reduced the total $H_2S$ and $NH_3$ emissions by 72 and 68%, respectively ($p = 0.001$), compared to ~99% of the powder ($p = 0.001$). The highest concentrations of $H_2S$ and $NH_3$ decreased from $48.1 \pm 4.8$ and $1810 \pm 850$ ppm to $20.8 \pm 2.95$ and $775 \pm 182$ ppm for pellets, and to $22.1 \pm 16.9$ and $40.3 \pm 57$ ppm for powder.

### 3.3. Economic Assessment of Granular and Non-Granular Organic Fertilizer Use

Different granular and non-granular organic fertilizers can contain different amounts of essential nutrients that are vital for the normal development and growth of plants. Among all the organic fertilizers evaluated, manure has the lowest percentage of nitrogen (N), phosphorus ($P_2O_5$) and potassium ($K_2O$). Pellets made from manure or meat and bone meal store several or several dozens of times more N, $P_2O_5$ and $K_2O$ elements. Figure 8 shows the comparative amounts of basic nutrient elements in granular and non-granular organic fertilizers, their monetary expression, fertilization rates, price and the total costs per hectare.

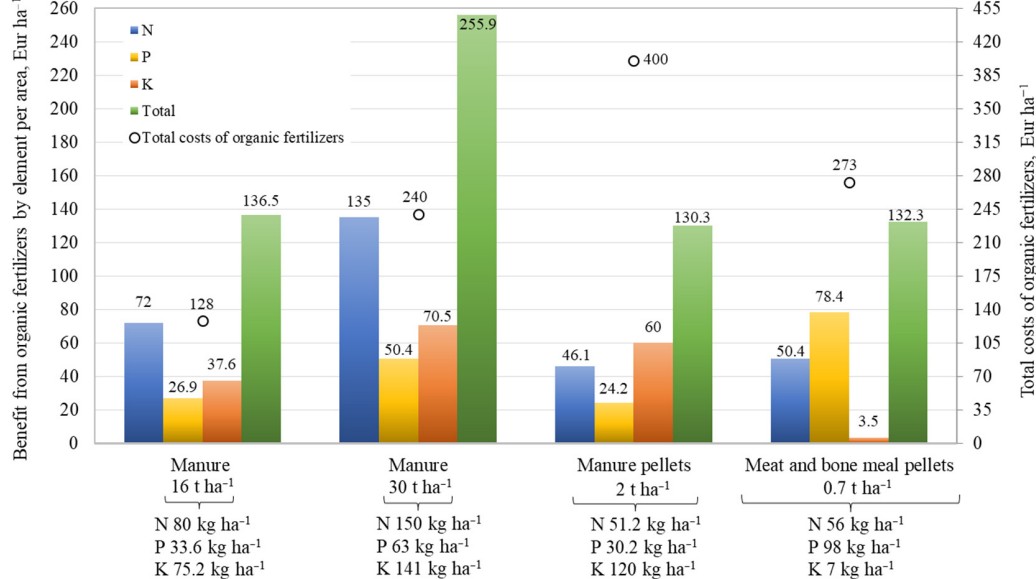

**Figure 8.** Nutrient content and economic indicators of granular and non-granular organic fertilizers (data were selected and calculated from the sources [11,27,32,37]).

Analysing the benefits of one tonne of organic fertilizers by individual nutrient elements, it was found that manure pellets and meat and bone meal pellets have significantly higher amounts of NPK elements than manure. The content of potassium in manure pellets and the content of nitrogen and phosphorus in meat and flour pellets are particularly distinguished. These results of the economic assessment show that the rate of manure pellets or meat and bone meal pellets can be many times lower than that of manure in order to incorporate similar amounts of nutrients into the soil. In addition, this is without assessing another advantage—that the pellets contain significantly more organic carbon, manganese, calcium and other elements. Meanwhile, the cost of organic fertilizers has shown that the cheapest organic fertilizer is manure, as it a livestock waste. However, it requires large areas of agricultural land [38], while manure pellets or meat and bone meal pellets have to pass a certain technological process until they take the form of pellets.

Another very important aspect of the cost-effectiveness assessment is the cost of mechanized technological operations of granular and non-granular organic fertilizers, such as loading, transportation and spreading in the field (Table 4).

**Table 4.** Costs of mechanized technological operations for the application of organic fertilizers (data were selected and calculated from the source [32]).

| Organic Fertilizers (Fertilization Rate) | Loading, EUR ha$^{-1}$ | Transportation, EUR ha$^{-1}$ | Spreading, EUR ha$^{-1}$ | Total Costs, EUR ha$^{-1}$ |
|---|---|---|---|---|
| Manure (16 t ha$^{-1}$) | 4.64 | 22.24 | 26.30 | 53.18 |
| Manure (30 t ha$^{-1}$) | 8.70 | 41.70 | 35.21 | 85.61 |
| Manure pellets (2.0 t ha$^{-1}$) | 0.58 | 2.78 | 3.29 | 6.65 |
| Meat and bone meal pellets (0.7 t ha$^{-1}$) | 0.20 | 0.97 | 1.15 | 2.32 |

Analysis of the costs of loading, transportation and spreading granular and non-granular organic fertilizers has shown that the highest costs per hectare are incurred when fertilizing with non-granular manure (Figure 9). Depending on the fertilization rate, 53.2 to 85.6 EUR ha$^{-1}$ is spent on these technological operations. This is because running one manure spreader is not enough to fertilize an area of one hectare. Depending on the capacity of the organic fertilizer spreader, it may be necessary to drive several times. Distance also has a big impact. The farther the fertilized field is from the livestock complex, the higher the transportation costs. In this study, the distance to the field was 7.5 km, which is a typical average transport distance for farms in the region. Increasing the productivity and sustainability of the farm requires the development of solutions to technological and logistical problems [1,21]. Fertilization with manure pellets with one fully loaded organic fertilizer spreader can fertilize only a few hectares, and fertilization with meat and bone meal pellets can fertilize several dozen hectares. This, of course, significantly reduces the costs.

Organic fertilizer pellets have a number of other advantages over manure that cannot always be monetized. Dry organic fertilizer pellets are much easier, cleaner and safer to transport than wet manure [39]. The high temperature of pellet production allows the removal of any harmful pathogens or microorganisms, so safe and nutritious fertilizers reach the soil and the plants. The granulation process also reduces unpleasant smells, which is very important when transporting and fertilizing fields [40], especially closer to settlements. Unlike manure, granular organic fertilizers can be stored and used as needed. There is a wider choice of agricultural machinery for spreading dry fertilizer pellets.

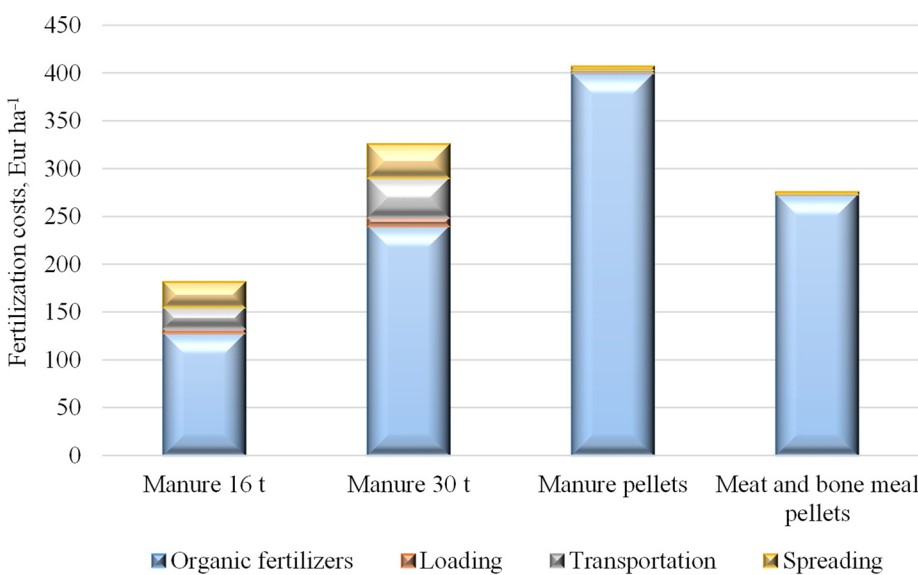

**Figure 9.** Total costs of fertilization with granular and non-granular organic fertilizers.

## 4. Discussion

The use of organic fertilizers is important, not only for enhancing soil productivity, but also for increasing the production of waste-free, more environmentally friendly agricultural products. The analysis of this study showed that energy, environmental and economic differences were obtained between the application of granular and non-granular organic fertilizers. Assessment of the energy consumption for mechanized technological operations showed that the energy consumption for non-granulated manure (16 and 30 t ha$^{-1}$) for machinery, diesel fuel and working time was several times higher than for manure pellets or meat and bone meal pellets. The fact that manure fertilization is very energy intensive is confirmed by the results of research presented in other sources. Reference [41] states that the highest share of energy consumption in organic beet cultivation was accounted for by manure costs, which varied from 48 to 53%, depending on the weed control method. The results of another study showed that farmyard manure-based energy consumption accounted for the largest share of 45.0–49.3% of total energy input in organic production of winter wheat and spring barley [42].

The results of environmental impact assessment show that the average NH$_3$ gas emissions from granular organic fertilizers was six times lower than from non-granular farmyard manure. Reference [43] states that the use of granular poultry manure reduces the risk of environmental pollution compared to non-granular fertilizers (manure) or mineral fertilizers. The properties of the treated manure made it possible to improve the nitrogen retention, and it is therefore advisable to increase farm-scale testing using pellets in order to assess the potential for large manure surfaces and the technoeconomic assessment [44]. The most common poultry manure management practices in Poland include the use of poultry manure on the soil (spreading unprocessed poultry manure on land, improvement of fertilizer and soil, various compositions and forms such as unprocessed poultry manure, granular forms, pellets and compost) through anaerobic digestion. Examples of soil use include: use of poultry manure as unprocessed organic fertilizer, raw material for composting with other agricultural residues, raw material for processing into granular and other fertilizers mixed with mulch and minerals (e.g., dolomite, lignite, peat). Ash from burning manure can be used as an additive to fertilizers [45–47].

Other studies have attempted to dry poultry manure with rice husks in the sun and turning them into pellets. Drying and grinding manure reduces unpleasant smells and is used as an SRF (slow-release fertilizer) which allows for longer keeping of nutrients [48,49].

Due to the segregation of crops and livestock, manure is often concentrated on a regional scale, which may result in more nutrients than the region needs. This can lead to

overuse of manure in the regions with concentrated cattle rearing, while crop-producing regions rely more on mineral fertilizers, leading to regional manure redistribution solutions to improve manure transportation. Manure recycling can be used to transport long distances and to separate P and N from each other and thus to improve their reuse. The purpose of processing is usually to reduce the mass of the manure and to concentrate the nutrients to improve their transportation. It is also important to produce fertilizer products that would replace mineral fertilizers and reduce environmental emissions. The market for new manure-based fertilizer products is still in its infancy, and storage and distribution practices still need to be developed [50].

Another 56 day incubation study was conducted to investigate the effects of alkaline organic residue pellets on: (1) the biological and chemical properties of acidic clay soils and (2) the emissions of carbon dioxide ($CO_2$), methane ($CH_4$) and nitrous oxide ($N_2O$). The study showed that alkaline residue-based organic pellets improved soil properties and can play an important role in reducing greenhouse gases in acidic clay soils. Therefore, the use of alkaline organic residue pellets in combination with chemical fertilizers can be considered a sustainable approach to agriculture [51]. Applications of organic fertilizers (OF) and chemical nitrogen (N) fertilizers (CF) as homogeneous granulation (evenly distributed in space) and spatial heterogeneous granulation (distributed separately), where N transformation processes (nitrogen oxide ($N_2O$) emissions) have a significant impact, makes the spatial distribution of OF and CF promising [52]. Economic assessment showed that the lowest costs for the purchase of organic fertilizers are incurred using manure. Depending on the fertilization rate, they are equal to 128–240 EUR ha$^{-1}$. The highest costs are incurred when using manure pellets (400 EUR ha$^{-1}$). The energy consumption for drying manure is about 100 kWh t$^{-1}$ of product [53]. As a result, manure pellets price per tonne is several times higher. Although the price of manure is lower, the fertilization with it generates lower yields as well [54]. Reference [43] states that the yields of spring wheat are 63% higher when organic granular fertilizers are used compared to non-granular fertilizers. Reference [55] found that the rice yield is about 7.21 t ha$^{-1}$ by fertilizing with organic granular fertilizers. However, the addition of mineral fertilizers to organic (green manure and farmyard manure) fertilizers showed a significant increased plant biomass production, which upon incorporation stimulates soil biological activity [56]. Using organic matter in plant fertilization increases the microbial/chemical actions and properties of soils [57]. A balanced application of fertilizers concludes to the formation of favorable conditions for the development of microorganisms, growth of plants and for an intense and lasting enzymatic activity [58].

However, when estimating the costs for mechanized technological operations of loading, transportation and spreading, the lowest costs are obtained by fertilizing with meat and bone meal pellets (2.3 EUR ha$^{-1}$) and manure pellets (6.7 EUR ha$^{-1}$). When fertilizing with manure, depending on the fertilization rate, the costs for these technological operations are several to dozens of times higher (53.2–85.6 EUR ha$^{-1}$). Assessing all aspects, the lowest total costs are obtained by fertilizing with non-granular manure (16 t ha$^{-1}$) and meat and bone meal pellets. However, organic fertilizer pellets have a number of other advantages over manure that cannot always be monetized. Furthermore, organic fertilizer pellets have a number of other advantages [39,40], one such advantage—granular fertilizers are more suitable for modern agricultural machinery used in precision farming.

## 5. Conclusions

Energy consumption of fertilization with organic fertilizer pellets for mechanized technological operations was 3.2 to 4.6 times lower compared to manure fertilization (30 t ha$^{-1}$). The most energy efficient is the use of meat and bone meal pellets, where the total energy consumption for fertilizers and mechanized technological operations is about 4136 MJ ha$^{-1}$, while for non-granular manure at fertilization rates of 16 and 30 t ha$^{-1}$, they account for 8156 and 14,843 MJ ha$^{-1}$, respectively.

Analysis of different organic fertilizers under the principle of laser spectroscopy revealed that granular organic fertilizers reduce the negative impact of $NH_3$ emissions on the environment by 97.8% compared to non-granular manure.

Assessing the different scenarios (manure 16 and 30 t; manure pellets, meat and bone meal pellets) of organic fertilizers, which allow to achieve similar benefit from organic fertilizers by nutrient element (N, $P_2O_5$, $K_2O$) per area (130.3–136.5 EUR $ha^{-1}$), the lowest total costs are obtained by fertilizing with non-granular manure (16 t $ha^{-1}$) and meat and bone meal pellets.

The results of the analysis allow estimating the energy and economic costs of fertilizing with granular and non-granular organic fertilizers. However, in order to determine the future effectiveness of organic fertilizers, it is necessary to carry out experimental studies of the effects of these fertilizers on soil and plants, as changes in soil properties and different crop yields can provide important information to support the effectiveness.

**Author Contributions:** Conceptualization, E.Š., V.N., Z.K., E.J. and A.J.; methodology, E.Š., V.N., Z.K.; validation, E.Š.; formal analysis, E.Š., V.N., K.L., Z.K., E.J., A.J. and R.Z.; investigation, E.Š., V.N., K.L., Z.K., E.J., A.J. and R.Z.; data curation, E.Š., V.N. and Z.K.; writing—original draft preparation, K.L.; writing—review and editing, E.Š., V.N., K.L., Z.K., E.J., A.J. and R.Z.; supervision, E.Š.; funding acquisition, E.Š. and A.J. All authors have read and agreed to the published version of the manuscript.

**Funding:** This research received no external funding.

**Institutional Review Board Statement:** Not applicable.

**Informed Consent Statement:** Not applicable.

**Data Availability Statement:** Not applicable.

**Conflicts of Interest:** The authors declare no conflict of interest.

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
