# Peer review of "Application of Granular and Non-Granular Organic Fertilizers in Terms of Energy, Environmental and Economic Efficiency"

_sustainability, doi:10.3390/su13179740_

Round 1
Reviewer 1 Report
The aim of this study was to evaluate the energy and economic efficiency of technological operations of manure pellets, meat and bone meal pellets and different rates of animal manure, as well as to investigate the impact of manure and meat and bone meal pellets on the environment. Authors have done a good job. However I have few suggestions regarding this manuscript:
Please highlight as better is possible the aim of the study (as a separate last paragraph of Introduction section). What makes special this study? Which is its novelty character or its special aspects? Why have the authors chosen this topic? What differentiate this paper from others in the same topic? Please make this aim of the study more relevant.
Figure 1 is blurred. Please replace it with a better quality one.
Please check and provide details for all apparatus used (model, manufacturer, city, country) and for chemicals/fertilisers (manufacturer, Country).
Please set the tables according to Instructions for authors Sustainability. https://www.mdpi.com/journal/sustainability/instructions
Discussion section is missing! It is extremely important to discuss and compare your results with others from already published papers. please add it and complete. Authors should discuss the results and how they can be interpreted in perspective of previous studies and of the working hypotheses. The findings and their implications should be discussed in the broadest context possible and limitations of the work highlighted. Part of actual Conclusion section must be moved here, as Conclusions are much too long. Also, I suggest few papers you may consult and refer to: Bungau S.; et al. Expatiating the impact of anthropogenic aspects and climatic factors on long term soil monitoring and management. Environ Sci. Pollut. Res. 2021, 30528-30550. https://doi.org/10.1007/s11356-021-14127-7 ; Samuel A.D.; et al. Effects of long term application of organic and mineral fertilizers on soil enzymes, Rev. Chim. 2018 , 69(10), 2608-2612. https://doi.org/10.37358/RC.18.10.6590 ; Samuel, A.D.; et al. Effects of liming and fertilization on the dehydrogenase and catalase activities, Rev. Chim. 2019 , 70(10), 2019, 3464-3468. https://doi.org/10.37358/RC.19.10.7576 , regarding fertilisers, importance, benefits, disadvantages, increasing productivity, etc., similarities with your results. Some data provided in the 3 papers above mentioned can be used and are very useful to improve the new Discussion section.
Conclusions section must briefly present the main findings of your work, highlighting them as better is possible.
Author Response
Point 1. Please highlight as better is possible the aim of the study (as a separate last paragraph of Introduction section). What makes special this study? Which is its novelty character or its special aspects? Why have the authors chosen this topic? What differentiate this paper from others in the same topic? Please make this aim of the study more relevant.
Response 1: Thank you for useful suggestions. The last paragraph of Introduction section was supplemented.
Point 2. Figure 1 is blurred. Please replace it with a better quality one.
Response 2: The reviewer's comment was taken into account and Figure 1 was corrected to better quality.
Point 3. Please check and provide details for all apparatus used (model, manufacturer, city, country) and for chemicals/fertilisers (manufacturer, Country).
Response 3: The reviewer's comment was taken into account and more detailed information on used apparatus was provided.
Point 4. Please set the tables according to Instructions for authors Sustainability. https://www.mdpi.com/journal/sustainability/instructions
Response 4: Tables were corrected.
Point 5. Discussion section is missing! It is extremely important to discuss and compare your results with others from already published papers. please add it and complete. Authors should discuss the results and how they can be interpreted in perspective of previous studies and of the working hypotheses. The findings and their implications should be discussed in the broadest context possible and limitations of the work highlighted. Part of actual Conclusion section must be moved here, as Conclusions are much too long. Also, I suggest few papers you may consult and refer to: Bungau S.; et al. Expatiating the impact of anthropogenic aspects and climatic factors on long term soil monitoring and management. Environ Sci. Pollut. Res. 2021, 30528-30550. https://doi.org/10.1007/s11356-021-14127-7 ; Samuel A.D.; et al. Effects of long term application of organic and mineral fertilizers on soil enzymes, Rev. Chim. 2018 , 69(10), 2608-2612. https://doi.org/10.37358/RC.18.10.6590 ; Samuel, A.D.; et al. Effects of liming and fertilization on the dehydrogenase and catalase activities, Rev. Chim. 2019 , 70(10), 2019, 3464-3468. https://doi.org/10.37358/RC.19.10.7576 , regarding fertilisers, importance, benefits, disadvantages, increasing productivity, etc., similarities with your results. Some data provided in the 3 papers above mentioned can be used and are very useful to improve the new Discussion section.
Response 5: Discussion section was added and supplemented by new literature sources.
Point 6. Conclusions section must briefly present the main findings of your work, highlighting them as better is possible.
Response 6: According reviewer comment conclusions was corrected and briefly present.
Reviewer 2 Report
On line 39, instead of writing "Stewart et al. [5] ", you can write" Ref. [5]… ”. The same goes for line 75 - instead of saying "Česonienė and Rutkovienė [23]" you can write "Ref. [23]… ”. You can also introduce these citation rules at the end of the article.
In lines 65-66 the Authors wrote "It is very important not to spread too much fertilizer, as improper use of fertilizers reduces the efficiency of plant nutrient use and ...". It could be expected that after such a sentence, the issue of precision farming would be developed, which would include adjusting the dose of fertilizers to the needs of plants. In my opinion, the Introduction should be supplemented with a paragraph concerning precision agriculture and its important element, i.e. precise fertilization.
I would like to ask on what basis the equipment of the given power (loader and tractor) and the fertilizer spreader with the given capacity were selected for the tests. Has the tractor been well matched (in terms of engine power) to the fertilizer spreader? Please let me know about it in the article.
In line 114, when it comes to energy consumption, it would be worth adding - in my opinion - that it is useful energy to distinguish this energy from recorded energy.
In line 115, in the fragment "... the energy equivalent of working time is 1.96 MJ h-1 ..." it is worth considering more precisely that it is about the time of human work.
In the Introduction, the Authors wrote about the doses of manure spread on the area of one hectare (lines: 42-43), hence the manure doses included in the research (line: 121) are justified. I think that in the Introduction one could also write about the doses of manure pellet fertilization and doses of fertilization with meat and bone meal pellets. Thanks to this, it would be easier in the research to justify the doses of these fertilizers (lines: 122-123).
Formula (1) on line 153 should be fully described with what the symbols mean in the formula. Under the formula, you need to write what the symbols mean and what their units are. In this way, it would be possible to check at the same time whether the formula is correct after compiling the units.
Should there be a dot at the end of formula (1)?
Table 1 shows “Manure pellet transportation and spreading” and determines “Field capacity”. The summary of these data shows that the transport has been combined with field capacity. Does transport count as field work? If transport is included in the calculation of the technological process, the distance to and from the field must be taken into account. This element determines the transport time and, in the next step, the efficiency of the manure spreading machine. In my opinion, the authors should calculate the efficiency of spreading manure (and other compared fertilizers) on the field, and exclude transport, possibly as a separate treatment (if it is needed for the analysis).
Table 1 includes "Manure loading", "Manure pellet loading" and "Loading of meat and bone meal pellets", specifying field capacity in [ha h-1] as the capacity unit. I have a question: Why is the loading capacity given in hectares per hour? When considering mass loading work, the performance of the device (loader) should be given in mass units per time unit. Therefore, I suggest adding one more column in Table 1 with the mass capacity of the loader. Theoretically, the efficiency of the loader can be calculated in [ha h-1], however, it requires taking into account the mass efficiency and the fertilizer dose. For the clarity of the presented results, however, it is better to specify the loader efficiency in [t h-1].
In Table 1 "Working time" is the reciprocal of "Field capacity", so theoretically it would be enough to specify one of these parameters if they relate to the operation of machines. If, on the other hand, "working time" refers to human work (which is probably worth considering in the analysis), then instead of the [h ha-1] unit, it would be worth taking into account the unit [m-h ha-1], i.e. man-hour per ha.
The information given in lines 223-225 is an exact repetition of the information presented in lines 114-116, even with the same citations, i.e. [31,32] (subsection: 2.1. Energy assessment of organic fertilizer use). In my opinion, there is no need to provide the same information (data) twice in different parts of the article. When presenting the research results, it could only be mentioned that the conversion factors given in the previous section were taken into account.
I do not know what is the difference between the sentence "Fertilization with manure (16 t ha-1) for mechanized operations was 636.4 MJ ha-1" (line: 228) from the sentence "The energy consumption of manure fertilization (16 t ha- 1) for mechanized operations was 636.4 MJ ha-1” (lines: 230-231)? In my opinion, one of these sentences is completely unnecessary, the more so are the sentences placed close to each other. The same remark applies to the sentence „Increasing the fertilization rate of non-granulated manure to 30 t ha-1 increased energy consumption by about 17%”, which is repeated twice in the same paragraph, on page 6.
In the case of Table 2, I think it would be worth choosing some concepts, which I wrote about earlier. It is a bit misleading when in the Table "Energy inputs" are given in [MJ ha-1] and the hectares in this unit are related to working time. In my opinion, in the article, instead of using the term "working time", the word "labor" would have to be used.
On lines 293-294 it says "... values in the range from 0 to 26000 min was observed", while in Figures 2 and 3 the scale ends at 24000 min. Perhaps it would be worth correcting this information and taking into account the information in the quoted part of the sentence: 24,000 min.
In Figures 2 and 3, on one of the axes the "Storage duration of manure pellets" is given in [min], while in Figure 4 the same value was considered in [h]. Perhaps it would be worth standardizing the units and using only one of them, because it would make it easier to analyze individual information.
In the computational analysis presented in Table 5, the transport distance was assumed to be 15 km (line 441). Is this a typical transport distance for farms in the region where the research was conducted?
Lines 486-488 read: "Assessing the three scenarios of organic fertilizers, which allow to achieve similar benefit from organic fertilizers by element per area…". I would like to ask, what „element” did the authors mean in writing this sentence?
Author Response
First of all, we would like to thank for additional comments, detailed recommendations and contributions to improving the quality of this manuscript. All of the issues raised in the Reviewers comments were corrected and are listed below. All revisions are highlighted in the text.
Point 1. On line 39, instead of writing "Stewart et al. [5] ", you can write" Ref. [5]… ”. The same goes for line 75 - instead of saying "Česonienė and Rutkovienė [23]" you can write "Ref. [23]… ”. You can also introduce these citation rules at the end of the article.
Response 1: Thank you, for the notice. We have changed the citation according your advice.
Point 2. In lines 65-66 the Authors wrote "It is very important not to spread too much fertilizer, as improper use of fertilizers reduces the efficiency of plant nutrient use and ...". It could be expected that after such a sentence, the issue of precision farming would be developed, which would include adjusting the dose of fertilizers to the needs of plants. In my opinion, the Introduction should be supplemented with a paragraph concerning precision agriculture and its important element, i.e. precise fertilization.
Response 2: We are very grateful for your comment and added information about precision farming as required.
Point 3. I would like to ask on what basis the equipment of the given power (loader and tractor) and the fertilizer spreader with the given capacity were selected for the tests. Has the tractor been well matched (in terms of engine power) to the fertilizer spreader? Please let me know about it in the article.
Response 3: These machines were matched to each other according to their power and operating parameters according to the manufacturer's recommendations and in accordance with the recommendations prepared by the Lithuanian Institute of Agrarian Economics [32]. The text has been supplemented.
Point 4. In line 114, when it comes to energy consumption, it would be worth adding - in my opinion - that it is useful energy to distinguish this energy from recorded energy.
Response 4: Thank You for the suggestion. We added that it is useful energy.
Point 5. In line 115, in the fragment "... the energy equivalent of working time is 1.96 MJ h-1 ..." it is worth considering more precisely that it is about the time of human work.
Response 5: Thank You for the clarification. The sentence was corrected.
Point 6. In the Introduction, the Authors wrote about the doses of manure spread on the area of one hectare (lines: 42-43), hence the manure doses included in the research (line: 121) are justified. I think that in the Introduction one could also write about the doses of manure pellet fertilization and doses of fertilization with meat and bone meal pellets. Thanks to this, it would be easier in the research to justify the doses of these fertilizers (lines: 122-123).
Response 6: The reviewer's comment was taken into account. The introduction was supplemented with doses of organic fertilizers used in this study.
Point 7. Formula (1) on line 153 should be fully described with what the symbols mean in the formula. Under the formula, you need to write what the symbols mean and what their units are. In this way, it would be possible to check at the same time whether the formula is correct after compiling the units.
Response 7: Thank you for useful observed. The inaccuracy has been corrected.
Point 8. Should there be a dot at the end of formula (1)?
Response 8: Dot is not necessary. It was deleted.
Point 9. Table 1 shows “Manure pellet transportation and spreading” and determines “Field capacity”. The summary of these data shows that the transport has been combined with field capacity. Does transport count as field work? If transport is included in the calculation of the technological process, the distance to and from the field must be taken into account. This element determines the transport time and, in the next step, the efficiency of the manure spreading machine. In my opinion, the authors should calculate the efficiency of spreading manure (and other compared fertilizers) on the field, and exclude transport, possibly as a separate treatment (if it is needed for the analysis).
Response 9: Thanks to the Reviewer for the comment. We can agree that the field capacity for transportation could be eliminated. However, it is very important to have full information in the analysis that allows to calculate the working time per hectare for all operations. If we take transport separately and spread fertilizer separately, the results would be completely different. For example, if manure were spread without stopping, the field capacity would be about 8.6 ha / h. However, working with one spreading machine, part of the time is spent for transportation organic fertilizers to the field (The distance from the field to the farm and back was 15 km). Therefore, these two technological operations were merged together. We have supplemented the text for clarity and removed Field capacity in Table 1 to avoid confusion.
Point 10. Table 1 includes "Manure loading", "Manure pellet loading" and "Loading of meat and bone meal pellets", specifying field capacity in [ha h-1] as the capacity unit. I have a question: Why is the loading capacity given in hectares per hour? When considering mass loading work, the performance of the device (loader) should be given in mass units per time unit. Therefore, I suggest adding one more column in Table 1 with the mass capacity of the loader. Theoretically, the efficiency of the loader can be calculated in [ha h-1], however, it requires taking into account the mass efficiency and the fertilizer dose. For the clarity of the presented results, however, it is better to specify the loader efficiency in [t h-1].
Response 10: The technological operation of loading organic fertilizers, like the spreading of fertilizers, cannot take place without stopping. When working with one spreading machine, loading only takes place for a certain part of the time, at another time the preparation for loading takes place. Therefore, the working time of the loader and the loader operator was also estimated per hectare. We have supplemented the text with loader efficiency (t / h) and removed Field capacity in Table 1 to avoid confusion as suggested by Reviewer.
Point 11. In Table 1 "Working time" is the reciprocal of "Field capacity", so theoretically it would be enough to specify one of these parameters if they relate to the operation of machines. If, on the other hand, "working time" refers to human work (which is probably worth considering in the analysis), then instead of the [h ha-1] unit, it would be worth taking into account the unit [m-h ha-1], i.e. man-hour per ha.
Response 11: We took note to the Reviewers comment to Table 2 below and changed term to "labor" instead of "working time". We believe that the term “labor” is more appropriate for performing analysis and evaluating human labor and machine work.
Point 12. The information given in lines 223-225 is an exact repetition of the information presented in lines 114-116, even with the same citations, i.e. [31,32] (subsection: 2.1. Energy assessment of organic fertilizer use). In my opinion, there is no need to provide the same information (data) twice in different parts of the article. When presenting the research results, it could only be mentioned that the conversion factors given in the previous section were taken into account.
Response 12: The sentence was rewrite according comment of Reviewer.
Point 13. I do not know what is the difference between the sentence "Fertilization with manure (16 t ha-1) for mechanized operations was 636.4 MJ ha-1" (line: 228) from the sentence "The energy consumption of manure fertilization (16 t ha- 1) for mechanized operations was 636.4 MJ ha-1” (lines: 230-231)? In my opinion, one of these sentences is completely unnecessary, the more so are the sentences placed close to each other. The same remark applies to the sentence „Increasing the fertilization rate of non-granulated manure to 30 t ha-1 increased energy consumption by about 17%”, which is repeated twice in the same paragraph, on page 6.
Response 13: Thanks to the Reviewer for the good attention. We revised this paragraph and deleted the repetition.
Point 14. In the case of Table 2, I think it would be worth choosing some concepts, which I wrote about earlier. It is a bit misleading when in the Table "Energy inputs" are given in [MJ ha-1] and the hectares in this unit are related to working time. In my opinion, in the article, instead of using the term "working time", the word "labor" would have to be used.
Response 14: The Reviewer's comment was taken into account and in Table 1 and Table 2 we changed "labor" instead of "working time". Hopefully this will be clearer as it applies to both human work and machine work.
Point 15. On lines 293-294 it says "... values in the range from 0 to 26000 min was observed", while in Figures 2 and 3 the scale ends at 24000 min. Perhaps it would be worth correcting this information and taking into account the information in the quoted part of the sentence: 24,000 min.
Response 15: Information corrected.
Point 16. In Figures 2 and 3, on one of the axes the "Storage duration of manure pellets" is given in [min], while in Figure 4 the same value was considered in [h]. Perhaps it would be worth standardizing the units and using only one of them, because it would make it easier to analyze individual information.
Response 16: In Figures axes the "Storage duration of manure pellets" values were considered in minutes, ours, days. It was for this purpose that different detailed graphs were presented. In this case, it is not treated as specific standard units, but for the periods during which the effect was determined, and for detailed analyzes it is useful to know in more detail and averages in minutes and generalized averages per hour and per day. For the practical application of every farmer, a thorough knowledge is more valuable than knowing the effectiveness of only the daily average without seeing variations. Therefore, it was specifically decided to show in detail and precisely for all the desired cases that both sudden instantaneous minutes and long-term effects during the days were seen.
Point 17. In the computational analysis presented in Table 5, the transport distance was assumed to be 15 km (line 441). Is this a typical transport distance for farms in the region where the research was conducted?
Response 17: The reviewer is right. This is a typical average transport distance for farms in the region. The distance from the field to the farm and back was 15 km (one direction 7.5 km). The text was supplemented.
Point 18. Lines 486-488 read: "Assessing the three scenarios of organic fertilizers, which allow to achieve similar benefit from organic fertilizers by element per area…". I would like to ask, what „element” did the authors mean in writing this sentence?
Response 18: Nutrient element: N, P2O5, K2O.
Reviewer 3 Report
This paper is unique one reporting organic fertilizers in terms of energy, environmental and economic efficiency and the results are interesting, too!
However, please add some information as below;
1) Page7 Figure 2.
You put an approximate formula "y=5E-06x+0.5087" on the figure.
Please add coefficient of determination "R2=0.XXXX"
2) Page 8 Figure 3.
You put an approximate formula "y=1E-0.6x+0.1246" on the figure.
Please add coefficient of determination "R2=0.XXXX"
3) Page 12 Table 4.
This table is important of this paper and it should show the data clearly.
So, please change this table to a figure to more understandable and comparable.
Author Response
First of all, we would like to thank for additional comments, detailed recommendations and contributions to improving the quality of this manuscript. All of the issues raised in the Reviewers comments were corrected and are listed below. All revisions are highlighted in the text.
Point 1. 1) Page7 Figure 2.
You put an approximate formula "y=5E-06x+0.5087" on the figure.
Please add coefficient of determination "R2=0.XXXX"
Response 1: According to Reviewer comment Figure were fulfilled and formulas was added with coefficient of determination.
Point 2. 2) Page 8 Figure 3.
You put an approximate formula "y=1E-0.6x+0.1246" on the figure.
Please add coefficient of determination "R2=0.XXXX"
Response 2: According to Reviewer comment Figure were fulfilled and formulas was added with coefficient of determination.
Point 3. Page 12 Table 4.
This table is important of this paper and it should show the data clearly.
So, please change this table to a figure to more understandable and comparable.
Response 3: Thank you for useful observed. Table was changed to figure.
Round 2
Reviewer 1 Report
The authors have addressed all my requests and did an impressive work.
2 minor corrections are needed:
- Please remove Ref. pink text from Introduction (L41, 71, 86)
- 2 empty Tables remained L299 and L417. Please remove them.
Reviewer 2 Report
Thank you for including the comments and suggestions presented in the review in the revised version of the article.